# Dynamic Outlier Detection in the Calibration by Comparison Method Applied to Strain Gauge Weight Sensors

**DOI:** 10.3390/s18124200

**Published:** 2018-11-30

**Authors:** Wojciech Walendziuk

**Affiliations:** Faculty of Electrical Engineering, Bialystok University of Technology, Wiejska 45D str., 15-351 Bialystok, Poland; w.walendziuk@pb.edu.pl; Tel.: +48-857-469-397

**Keywords:** metrology, calibration, comparison, force measurement, force sensor, weight sensor, strain gauge, dynamic time warping

## Abstract

The present work proposes a robust method of analyzing sets of data series shifted in time in respect to each other utilizing the process of dynamic calibration by comparison. Usually the Pearson’s correlation analysis coefficient is applied for this purpose. However, in some cases the method does not bring satisfactory results, as it can be seen in the results of the research conducted for the purpose of this paper. The Dynamic Time Warping method may be the solution to this problem, as it appears to be more efficient while comparing the shapes of calibration characteristics done with the use of the Pearson’s method. The presented method may also be applied to eliminate dynamic outliers collected in the process of recurrence examination or the analysis of strain gauge weight sensors hysteresis. This fact also makes the method a good tool for eliminating improper data series which might appear in the calibration process due to, e.g., malfunctioning devices installed in the calibration stand. The article presents an example of using the proposed method in eliminating improper dynamic characteristics obtained in a simulated calibration stand. Moreover, a comparative analysis performed on the simulation data is also presented in the article, as well as the result of the laboratory experiment.

## 1. Introduction

Constructing an electronic measurement device, equipped with signal conditioners, usually finishes with the calibration process. This is related to the fact that the end-user would like to conduct measurements with a device of a known accuracy, calibrated in a particular measurement unit and fulfilling the requirements of the standards determining parameters for devices, valued in the areas where they are exploited. The calibration process mentioned above applies mostly to measurement devices undergoing verification and approved for use in everyday life and industrial, medical or military purposes [1]. It is worth stressing that the regularly conducted calibration is one of the ways of providing adequate accuracy to a device. The relevant provisions regulating the methods of calibrating weighing devices, which will be discussed further, are included in recommendations of the following organizations: EURAMET—The European Association of National Metrology Institutes Calibration Guide [2], OIML—International Organization of Legal Metrology [3], or NIST—National Institute of Standards and Technology [4]. It must be mentioned that, the process of calibrating weighing devices is mainly based on research conducted as a static process by placing appropriate loads on force transducers [2,3,5,6].

This work presents an example method which may be applied in one of the calibration stages of devices in which comparing a set of characteristics obtained during dynamic tests is vital. As a result of such research, time series showing characteristics to be compared are obtained. Comparison by Pearson’s correlation coefficient method is usually used for this purpose. However, this method is not robust in factors such as time series shift, which makes the results unreliable. Therefore, the author of this work proposes to use the DTW (Dynamic Time Warping) method in order to eliminate the inconveniences. It allows conducting time characteristics comparative tests of a model device and the tested machine within a time unit determined in the process of calibration. Another advantage of using this solution may be effective elimination of incorrectly conducted experiments, which may derive from such conditions as apparatus imperfectness, the human factor, or the environment in which the tests are completed. This very aspect will be closely investigated in the present work.

The rest of this paper is organized according to the following structure. Firstly, the methodology of weighing devices calibration and the DTW method applied in the dynamic calibration are described. Then, the concept of an algorithm robust to the lack of time flow synchronization, being a result of comparative calibration tests, will be presented. The last part of the article will contain the results of simulations and experimental research conducted with the use of a measurement device equipped with a strain gauge transducer.

## 2. Weighting Instrument Calibration Procedure

Calibration tests of a newly-built weighing instrument are based on its comparison to the reference model. Usually, only the procedure of static tests is conducted. In this case, a set of actions defined within certain norms are executed. In the following stage, dynamic tests comparing, for example, the response of a weight transducer to an excitation in the form of actuating the transducer to sinusoidal vibration may be completed [7,8,9,10,11]. In some cases, it is necessary to compare the time series characteristics of the model and the tested transducer, which in the present work are hereinafter referred to as “dynamic characteristics”.

It is crucial to highlight that in all stages of the calibration process (Figure 1), such parameters as temperature, pressure, and relative air humidity must be kept constant.

During the whole process, a precise measure of the loads, calibrated periodically and verified according to the binding standards [12] on the tested device must be applied at all times. The calibration procedure may be then conducted as follows. With the mentioned conditions maintained, a repeatability test, which consists of repeatedly putting the same load on the strain gauge sensor, should be conducted. These procedures are followed by a test of errors of indication, a procedure performed with the use of *k* ≥ 5 different testing loads for previously determined testing points of the scales. The aim of this test is to evaluate the whole range of the device operation. In the next test (Figure 2), the testing load is placed in the areas of the strain gauge sensors and in the middle of the mass of the weighing instrument [13]. It is executed by putting a load (Figure 2) on the device: ① in the center; ② forward on the left; ③ backward on the left; ④ backward on the right; and ⑤ forward on the right. During tests, the load is first placed in position ①, then it is shifted into the other four positions. At the end of the test, the load is placed in position ① again.

It is crucial that whenever the load is removed, the scales should display the value of 0. The last stage includes subsidiary tests such as: linearity test, hysteresis test, tare test, time indication steadiness test, magnetic field influence test, and minimum mass test, which is used to define the smallest load which may be applied [13,14].

The static calibration process of a weighing instrument, mentioned above, is usually used with regard to the mentioned recommendations of normalization institutions. Nevertheless, sometimes it may be necessary to compare the work of the model and the tested devices in the dynamic state [8,15,16]. Strain gauge calibration used, for example, in examining patients’ stability, in weighing flowmeters or in systems measuring dynamic deflections and strain [17] can be used as examples. During such comparison, errors may appear in the transducers work which can be preliminarily eliminated during the static calibration. However, sometimes, while conducting a comparison of the time series characteristics, wrongly performed experiments may cause inconsistency of the obtained results of comparing, for example, 10 tests. It might appear that measurements conducted during such a series of tests may differ from each other, and the person conducting the comparison may find elimination of the wrong test difficult.

Pearson’s correlation method appears to be of great use here by eliminating the time series characteristics which are not coherent. However, it is not effective in the cases when the time series are of different lengths, are shifted with respect to each other, are not synchronized with each other in time (e.g., in the cases of uniform time scaling or time warping) or do not have identical amplitudes [18,19]. For example, in a dynamic test of a weighing transducer, at a three times repeated experiment, three characteristics of the dependence of mass increase in time will be obtained. While the characteristics are almost identical, but shifted with respect to each other in time, the linear correlation method will not be effective and, therefore, the result of the comparison will not give a reliable response if any of the characteristics diverge.

It should be stressed that while estimating the uncertainty of a single measurement result in static tests [20,21] there is a tool which enables eliminating the outlier samples with the use of statistical methods [22,23]. However, the mentioned methods can be applied only under condition that the samples are normally distributed [24,25,26]. Unfortunately, there is not a tool robust to the mentioned lack of synchronization applicable for dynamic measurements.

## 3. Application of the DTW to the Calibration by the Comparison

### 3.1. General Description of the DTW Method

The DTW method is applied in the majority of the cases for identifying and classifying time series used in sound recognition [27] or image recognition [28]. Data analysis in satellite imaging, called Satellite Image Time Series, is one of the interesting applications of this method. Using this method is based on comparing data randomly obtained from satellite images which, as a result of for example clouding, were not possible to be taken above the analyzed area of the Earth [29]. The subsequent stored satellite images are then compared with the use of the DTW method after appropriate decomposition of multidimensional multispectral matrices into a set of one dimensional sequences. In other works, the author presents their research on a vehicle velocity evaluating algorithm applied in a measurement system based on a wireless network of magnetic sensors. The applied algorithms calculating similarities of two magnetic signatures are mainly based on comparing the Euclidian distance function, or on applying a cross-correlation method [30,31]. Recently, some unusual but interesting applications of this method have appeared. One is used to analyze the time series of quaternions and the method has been called quaternion dynamic time warping (QDTW) [32].

Below we present how the DTW method can be applied for the analysis of data obtained in the dynamic calibration of weighing sensors, which for the purposes of this work has been called CCDTW (*Comparison/Calibration Dynamic Time Warping*).

The DTW algorithm compares series of data which are divergent in time and amplitude domains but have common characteristics (Figure 3). Using the comparison measure based on estimating the Euclidian distance in the element-wise Euclidean distance system (one by one element), would not bring positive results in this case because of the previously mentioned divergence factors resulting from the shift of time series with respect to each other [33].

A modified algorithm of characteristics comparison in time consists in using two signals: the model one *X_ref_* (1) and the measured one *X_meas_* (2), in which certain samples are defined as follows:(1)$$\left\{n_{ref}=1,\;2,\;3,\;\ldots ,\;N_{ref}\right\}\in X_{ref},$$
(2)$$\left\{n_{meas}=1,\;2,\;3,\;\ldots ,\;N_{meas}\right\}\in X_{meas},$$
where: *n_ref_*—representation of a single sample of the reference data vector and *n_meas_*—representation of the remaining data (obtained during comparative tests) vector.

The above data vectors are inserted in a matrix (Figure 4) and constitute two edges limits.

Afterwards, distances between the compared data included in the vectors should be defined with the use of, for example, the Euclidian distance (3). Calculated in this way element-wise values will form a matrix showing Euclidian dependences *X_dist_* between the model *X_ref_* and the measured *X_meas_* signals:(3)$$X_{dist}\left(n_{ref},n_{meas}\right)=\sqrt{\overset{N_{ref}\times N_{meas}}{\underset{i=1}{\sum }}{\left(X_{ref}\left(i\right)-X_{meas}\left(i\right)\right)}^{2}}.$$

It should be mentioned that the smallest distance values of the data obtained this way may mean that those points might be matched with each other. Calculating the cumulated cost *X_acc_* on the matrix edge (Figure 4) in relation to *X_ref_* (4) and *X_meas_* (5) will be the next step of the procedure. The following dependencies are obtained as a result:(4)$$X_{acc}\left(n_{ref},n_{meas}\right)=\overset{N_{ref}}{\underset{k=1}{\sum }}X_{dist}\left(n_{ref},1\right),$$
(5)$$X_{acc}\left(n_{ref},n_{meas}\right)=\overset{N_{meas}}{\underset{k=1}{\sum }}X_{dist}\left(1,n_{meas}\right),$$
and in the resulting area (6):(6)$$X_{acc}\left(n_{ref},n_{meas}\right)=\overset{N_{ref}\times N_{meas}}{\underset{k=2}{\sum }}X_{dist}\left(n_{ref},n_{meas}\right)\;.$$

As can be observed in Figure 5, the data placed diagonally contains small values corresponding with small distances between points *n_ref_*, *n_meas_*. Therefore, in order to determine the measure of similarity between the two signals, a so called accumulated cost path should be found in the above matrix.

In order to achieve the goal, the following assumptions should be taken for the analyzed data:point (1, 1) is the origin of creating the path, and (*N_ref_*, *N_meas_*)—its end;we can shift by one step preserving the condition of function continuity and monotonicity;the path will only run forward as particular samples are taken in time which we cannot reverse.

Taking into account the algorithm, the assumptions that are followed indicate the direction of searching the shortest path, which means that for point (*n_ref_*, *n_meas_*) we can move along diagonally (*n_ref_* + 1, *n_meas_* + 1), to the left—(*n_ref_* + 1, *n_meas_*), or upward—(*n_ref_*, *n_meas_* + 1). On this basis, Formula (7) can be formed:(7)$$X_{acc}\left(n_{ref},n_{meas}\right)=\min\{\begin{array}{ll} X_{acc}\left(n_{ref}-1,n_{meas}-1\right)+X_{dist}\left(n_{ref},n_{meas}\right) & -\mathrm{diagonal}\;\mathrm{step}\\ X_{acc}\left(n_{ref},n_{meas}-1\right)+X_{dist}\left(n_{ref},n_{meas}\right) & -\mathrm{right}\;\mathrm{step}\;\\ X_{acc}\left(n_{ref}-1,n_{meas}\right)+X_{dist}\left(n_{ref},n_{meas}\right) & -\mathrm{top}\;\mathrm{step}\; \end{array}$$

Next, a path designing the shortest way should be determined. The backtracking procedure, which consists of moving from point (*N_ref_*, *N_meas_*) to (1, 1), is applied for this purpose. The transition function is a crucial element of this stage of the procedure. The function finds the path of the least cumulative cost (*x_cc_*) in the matrix. The function describing this situation can be formed as (8):(8)$$x_{cc}=\overset{1}{\underset{i=N_{ref}}{\sum }}\overset{1}{\underset{j=N_{meas}}{\sum }}min\{\begin{array}{c} X_{acc}\left(i-1,j-1\right)\\ X_{acc}\left(i,j-1\right)\\ X_{acc}\left(i-1,j\right) \end{array}$$

After finding the shortest path, the elements that have been qualified as smallest are summed. This way we obtain the measure of comparison which may be applied in further characteristics compared in the process of a device calibration.

### 3.2. Reference Time Series Generation

A key question concerns the approach to construct a benchmark which will enable applying the proposed method. Such a benchmark (9) can be constructed by means of approximating all *n* elements taking part in the waveform analysis. (9)$$X_{ref}\left(i\right)=\frac{\sum_{i=1}^{N_{ref}}X_{meas}\left(i\right)}{n}$$

It should be stated that the formula presented above is only correct when the analyzed data series, for example from 10 experiments, are of the same length. In the case when the lengths are different, the measurement data vectors should be complemented with neutral data, placed on the ends or origins of the series. The neutral data may take on the values of the steady state that exists before executing the dynamic experiment.

Finally, after completing this procedure of data generation a matrix from which the reference model is generated, is obtained. It is worth stressing that such a model is a good reference providing that the incorrect dynamic characteristics do not overbalance the experiment. Should this be the case, the excess of incorrect data will result in constructing a model which in some cases may affect the effectiveness of the applied method. 

## 4. Simulation and Experimental Research

In order to check the CCDTW algorithm work, a set of simulation tests and experiments were executed. The experimental tests were carried out with the use of a device, whose function is to measure the flow of liquids, which is equipped with a weighing transducer. Previous experience gained during conducting research on signal conditioners [34,35] for devices [17,36], which may be used in non-invasive examinations of urine flow in patients with suspected BPH (*Benign prostatic hyperplasia*) [37,38,39], was used in this investigation.

It must be mentioned that authors of various publications frequently use the notion of volume flow rate—*Q* (m^3^/s) (10), which means the volume of the liquid flowing in a time unit:(10)$$Q=\frac{\underset{\Delta t\to 0}{lim}\Delta V}{\Delta t}=\frac{dV}{dt}$$ where: *V*—volume of fluid (m^3^) and *t*—time unit (s).

In the cases based on weighing transducers, which measure liquids flow (e.g., in uroflowmetry), the above measure is used, showing the flow in units (mL/s, or mL/s) [37,38,39]. To be more precise, considering for example the case of the construction of a device for examining the flow speed of liquids of different properties, Formula (11) should be applied. It allows converting the value of the volumetric flow rate to the mass flow rate expressed in (kg/s). (11)$$\overset{´}{m}=\frac{\underset{\Delta t\to 0}{lim}\Delta m}{\Delta t}=\frac{dm}{dt}=\rho \cdot Q=\rho \frac{dV}{dt}$$
where: *m*—mass (kg), ρ—mass density of the fluid (kg/m^3^), and *t*—time unit (s).

### 4.1. Test Bench Used in the CCDTW Evaluation

In order to examine the correctness of the CCDTW algorithm work, a laboratory stand consisting of appropriate functional modules was used (Figure 6). A module dosing the liquid (water in this experiment) is the first element of the equipment. It consisted of a container and an electromagnetic valve which initiates the liquid flow. The valve activation is executed with the use of a microprocessor circuit which precisely measures the time between its opening and closing. The time of the experiment was 10 s.

The IFM Electronic SM6000 magnetic-inductive flowmeter, whose basic properties are presented in Table 1, is another module of the system. The flowmeter is connected to a 16-bit measurement card and a PC by USB serial bus. The computer is equipped with a program created in National Instruments LabVIEW. The aim of this part of the system is to activate the measurement data collected on the flowmeter, which will further be used to check the flow value recorded by the machine.

The uroflowmeter is the last element of the equipment. It works in the mode that enables independent data recording. Its indications will be analyzed with respect to repeatability of the recorded data obtained during the experiment.

The meter that was used consisted of an electronic dynamometer and a mass sensor. During data recording, the actual results of mass measurement are displayed on an liquid-crystal display (LCD) and stored in non-volatile memory. The apparatus has an embedded real time clock and it can store up to 10 experiments, lasting up to 3 min each. The meter is equipped with an RS232 series communication connection that enables sending the measurements results to the computer and calibrating the device. In Table 2, the uncertainty budget for the constructed device is featured [36]. Below (Figure 6), a diagram showing the main elements of the used stand for examining the flow with the use of a uroflowmeter, is presented.

Using the presented laboratory stand to conduct the research resulted in the fact that all the measurements were completed in an identical way (repeatably).

Moreover, it was possible to apply identical measure of the liquid, which was dosed with the use of a microprocessor controller. Eventually, in order to test the CCDTW method, 10 measurement experiments, which were recorded and analyzed, were conducted.

### 4.2. Data Acquisition

The first stage of the research consisted in numerous tests of the work correctness of the measurement laboratory stand. During such a test, among others, the amount of the dosed fluid was checked. The fluid was poured from the container placed at the top of the stand to a vessel located on a weighing transducer being an element of the tested device. This procedure was done repeatedly, checking at the same time whether any significant discrepancies in the tests results were revealed. On this basis, the repeatability of the experiments was confirmed.

Conducting a set of measurements was the following stage of the research. They were recorded with the use of the equipment presented in (Figure 6). Ten subsequent dynamic characteristics were extracted to be analyzed.

### 4.3. Analysis Procedure

The stored data was analyzed using the algorithm presented in (Figure 7). The first stage consisted of importing the measurement or simulation data into the program memory. Then the data was allocated in a table containing *n* + 1 columns of data. The last column was dedicated to the benchmark which was generated during the experiment (Equation (9)). The benchmark contained approximated data from respective characteristics in the element-wise form, which is broadly discussed in Section 3.2 Reference time series generation. Then the operation of data comparison with the use of the CCDTW method and the Pearson’s correlation coefficient was executed. In order to do that, subsequent columns of data were compared with the model.

Formula (12) has been assumed as a measure of the Pearson’s correlation coefficient. It was used to determine the correlation coefficient for particular measurement series related to the mean value calculated from all series. (12)$$r\left(n_{ref},n_{meas}\right)=\frac{\sum_{i=1}^{N}\left(X_{ref}\left(i\right)-{\bar{X}}_{ref}\right)\left(X_{meas}\left(i\right)-{\bar{X}}_{meas}\right)}{\sqrt{\sum_{i=1}^{N}{\left(X_{ref}\left(i\right)-{\bar{X}}_{ref}\right)}^{2}}\sqrt{\sum_{i=1}^{N}{\left(X_{meas}\left(i\right)-{\bar{X}}_{meas}\right)}^{2}}}$$
where: *N* = *N_ref_* = *N_meas_*—normalized length of the data vectors.

## 5. Results of Method Analysis

A set of experiments showing the advantages of the CCDTW method was conducted with the use of the research procedure presented in the previous part of the paper. It was decided to apply an estimator of the Pearson’s correlation coefficient, for which a conversion range was used (13) and an appropriate amplification coefficient *r_ampl_* was added. (13)$$\hat{r}\left(n_{ref},n_{meas}\right)=\left(1-r\left(n_{ref},n_{meas}\right)\right)\times r_{ampl}$$
where: *r_ampl_*—scalar coefficient determining the amplification of the value of the estimated converted Pearson’s correlation coefficient.

It is worth noticing that the *r_ampl_* coefficient was set manually, in a way that enables presenting the result of comparing the CCDTW and the Pearson’s correlation methods in one diagram so that it is easy to analyze. Using the *r_ampl_* scale coefficient allowed the observation of the obtained results of comparisons through adjusting the correlation coefficient amplitude to the result of implementing the CCDTW method. Due to this, the coefficient value, enlarged in some analyzed cases, better exposed the relations between the compared methods. 

### 5.1. Simulation Results of Comparison of Pearson’s Correlation Coefficient Method vs. Calibration-by-Comparison Dynamic Time Warping for Ideal Time Series Without Disturbances

Below (in Figure 8, Figure 9, Figure 10, Figure 11, Figure 12, Figure 13 and Figure 14) we can see the results of the simulations of dynamic characteristics generated artificially. All characteristics are identical with respect to the amplitude, time of duration, and the number of samples. Subsequent diagrams show the result of the experiment connected with the № 1 dynamic characteristics shift within the range of <−15, +15> samples with the step of five samples.

As it is clearly observable, the result of the experiment proved the lack of effectiveness of the Pearson’s correlation method in relation to the CCDTW method. Although all characteristics were identical, the Pearson’s method appeared to have insufficient linear correlation. Whereas the CCDTW method proved to be robust to such discrepancies. Obviously, when identical characteristics coincided in time (Figure 11), both methods showed the lack of differences.

It should be mentioned that the result of the experiment showed a significant sensitivity if the Pearson’s correlation method to the time series shift towards one another, which is an undesired quality in this case. Additionally, it should be noticed that the described phenomenon appeared when the model characteristics № 11, generated with the use of the element-wise average, was not significantly deflected in relation to the other characteristics № 1–№ 10. 

### 5.2. Simulation Results of Comparison of Pearson’s Correlation Coefficient Method vs. Calibration-by-Comparison Dynamic Time Warping for Ideal Time Series with One Disturbed Series

The next experiment (Figure 15, Figure 16, Figure 17, Figure 18, Figure 19, Figure 20 and Figure 21) consisted in comparing artificially generated characteristics, where № 1 data series was shifted as in the previous test, but was additionally deflected. The deflection was of the sinusoidal shape, the duration of the added impulse equaled 10% of the dynamic characteristics and the amplitude equaled 30% of the maximum value of the ideal characteristics.

Similarly, as in the previous experiment, the CCDTW method proved to be most efficient for detecting incorrect characteristics. For large shifts (−15 and +15 samples), the correlation coefficient method showed incorrect characteristics with a similar efficiency. This was, however, caused by the fact that the Pearson’s method is not robust to the lack of synchronization of the analyzed characteristics in time. It can be clearly observed that for characteristics № 1, the Pearson’s correlation coefficient value is almost identical with the value obtained in the previous test (Section 5.1, ideal time series without disturbances, Figure 14). On this basis, we can conclude that disturbing the characteristics with a sinusoidal impulse did not significantly affect the process of identifying the incorrect time series with the Pearson’s correlation method.

### 5.3. Simulation Results of Comparing the Pearson’s Correlation Coefficient Method and the Calibration-by-Comparison Dynamic Time Warping for an Ideal Time Series with Two Disturbed Series

Another experiment was conducted on artificially generated data applying a more complex combination of time series shifting. In the first case (Figure 22), two characteristics (№ 2 and № 8) were deflected. Characteristics № 2 was deflected with an inverted sinusoidal impulse of the length of 10% of the dynamic characteristics and the amplitude of 8% of the model characteristics maximum value. Characteristics № 8 had a deflection of a sinusoidal wave of the length of 10% of the dynamic characteristics and the amplitude of 5% of the model characteristics maximum value. All the time series in this experiment are synchronized, not shifted in time with respect to each other.

For the analysis of this case, the *r_ampl_* coefficient has been adjusted manually so that it would be possible to show the effectiveness of detecting outliers. On the basis of the mentioned modification, it can be stated that the Pearson’s correlation coefficient method is less sensitive to detecting outliers (the *r_ampl_* coefficient was manually increased up to the value of 100), but equally effective as the CCDTW method.

In the following experiment (Figure 23), it was decided to shift three not deflected characteristics with the deflected ones remaining in the same position as in the previous case. Model characteristics were shifted as follows: № 1—shifted by 10 samples forward, № 3—shifted by five samples forward and № 10—shifted by five samples backward in relation to the neutral position. The term of neutral position relates to all characteristics synchronized with one another in time.

As it can be clearly observed, also in this case the Pearson’s correlation method identified time characteristics incorrectly. Although characteristics № 1, 3, and 10 were not deflected but shifted in relation to the neutral position, the linear correlation algorithm proved that they are divergent. Additionally, the linear correlation method did not show incorrect characteristics № 2 and № 8.

In the following test (Figure 24), an experiment for not shifted model characteristics was conducted. However, characteristics № 2 was shifted by five samples forward (time series accelerated in relation to other not deflected characteristics), and characteristics № 8 shifted by five samples backwards (delayed characteristics).

The analysis of this case proves the effectiveness of both methods. Characteristics are equally deflected, as it was previously, but they are shifted in respect to one another of a small number of samples. It is clearly observable that the result of applying both methods is directly connected with the method of generating the reference characteristics. It is the element-wise average in the analyzed case.

In the last test (Figure 25), not deflected characteristics № 3 and № 7 were additionally shifted. The data from characteristics № 3 was shifted by 10 samples forward, and the data from characteristics № 7—by five samples forward.

Analyzing the obtained results, it can be concluded that the CCDTW method proved to be more effective in detecting outliers № 2 and № 8. The Pearson’s correlation method detected the incorrect characteristics № 8. However, characteristics № 3 was detected as incorrect, although it was the model not deflected characteristics but shifted by 10 samples forward.

### 5.4. Real Experiment of the Weight Sensor Data Comparison of Pearson’s Correlation Coefficient Method vs. Calibration-by-Comparison Dynamic Time Warping

A further experiment consisted in recording the dynamic characteristics with the use of a weighing transducer (Figure 6). 10 subsequently recorded characteristics were analyzed for this purpose (Figure 26).

As a result, characteristics № 3 and № 9 were indicated as most outlying from the model one. If such cases are noticed in the process of a device calibration, a deeper insight in the reason of the differences should be provided.

### 5.5. Simulation Results Based on Real Experiment of the Weight Sensor Data Comparison of Pearson’s Correlation Coefficient Method vs. Calibration-by-Comparison Dynamic Time Warping

As the last experiment, the characteristics obtained during tests made for a weighing transducer but artificially deflected were compared. This deflection was applied for characteristics № 3 and № 8, which are marked in red. In the first case (Figure 27), the characteristics were synchronized with each other, in the second case (Figure 28 and Figure 29), characteristics № 3 was shifted by 15 samples forward (+15 samples), and characteristics № 8—by 15 samples backwards (−15 samples).

In the last case (Figure 30) it was decided to perform a very complex situation where, apart from testing deflected and shifted characteristics № 3, № 8, some characteristics were desynchronized: № 1—(+10 samples), № 2—(+15 samples), № 5—(−5 samples), and № 6—(−10 samples).

In this case, the use of the CCDTW method resulted in an effective outcome. The Pearson’s correlation method indicated incorrectness in particular characteristics as well, but less effectively. While comparing the distorted characteristics (Figure 27 and Figure 28), but not shifted in time, the increase of the *r_ampl_* coefficient value up of 100 allowed the mentioned incorrectness to be noticed. While in the case of № 3 and № 8 charts (Figure 29), additionally shifted in time, both methods appeared to be similarly efficient.

In the case presented in Figure 28, where characteristics № 3 and № 8 were additionally shifted in time, both methods proved to be similarly effective. This was caused by a significant deflection of the model characteristics whose element-wise average considerably differed from an average correct characteristics. At even greater deflection and desynchronizing the characteristics in time (Figure 30), the Pearson’s correlation method did not clearly show the outliers. But the CCDTW method indicated the deflected characteristics correctly. It should be added that in real calibration tests such great deflections and shifts in time are uncommon. However they were presented because of the possibility of applying the CCDTW method in comparing tests of dynamic characteristics not necessarily related to calibration of sensors or other measurement equipment.

## 6. Discussion

The Calibration-by-Comparison Dynamic Time Warping algorithm, presented in this paper, is an effective tool for comparing dynamic data series recorded during sensors calibration. The author has not come across any articles discussing such an application of the presented algorithms used to eliminate incorrect data series obtained during dynamic calibration of devices conducted by comparing a tested device with a model one. Moreover, the paper presents an algorithm of the DTW method in which a model generated by determining the mean value of the characteristics obtained during the calibration process. On this basis, simulations on artificially generated data and laboratory tests on the results obtained in an experiment were conducted. As a result, characteristics displaying the efficiency of the CCDTW method in comparison to the Pearson’s correlation coefficient was created. It is worth noticing that the present work does not present the results of the comparison of two measurement devices, but focuses on the method of eliminating data significantly different from the generated model. This resulted from the fact that the SM6000 sensor cannot be treated as a signal model, but only as an indicator. Therefore, further investigation will be connected with examining another application of the method of comparing the model device with the tested one. It is also planned to apply this solution in other types of sensors (not necessarily weighing sensors).

Analyzing the advantages and disadvantages of the CCDTW algorithm, the following conclusions may be stated. Among the positive aspects of the method, its robustness to the lack of synchronization of the compared measurement data in time or a simple way of determining the model data series by the element-wise average may be listed. Moreover, the algorithm itself is easily implemented in any programming environment. The drawbacks include the sensitivity to the shape of the model or the necessity of using data series of the same length. It is also troublesome to interpret the calculation results of the so-called cumulated transition cost represented in the form of scalar value. The value of this coefficient is difficult to attribute to the measure determining to what extent the characteristics are similar to each other. This latter disadvantage may, however, be attributed to the Person’s correlation method, discussed in this paper, in which the correlation result can be interpreted in different ways. In both methods, in spite of the differences in the principle of working, the result is treated arbitrarily, without giving a particular value which could be meant as the absolute measure.

Finally, also other possibilities of applying the CCDTW method are worth mentioning. It can be used for analyzing data recorded in a calibration stand during examining sensors in the oscillation cycle. Such examinations are applied to force sensors, on which loads are placed cyclically. In such a case it is possible to examine the dynamic response or the hysteresis loop effect of force sensors. The data recorded in such research could be compared to each other after previously dividing the dynamic characteristics in the points of the lack of load (steady state). Then the data should be complemented in a way to enable obtaining data series of the same length. The process would end according to the procedure presented in Section 4.3. Analysis procedure.

The author hopes that the presented method will become the subject of wide discussion concerning the possibilities of improving it. The main problem which needs to be solved is establishing new methods of generating the model characteristics. Moreover some additional analysis of the amplitude change influence of the compared signals on the final result of comparison would be desirable.

## Figures and Tables

**Figure 1 sensors-18-04200-f001:**
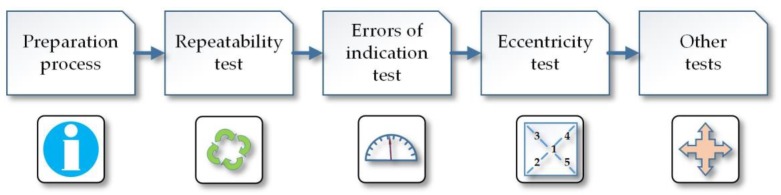
Simplified static calibration process of weighing instruments.

**Figure 2 sensors-18-04200-f002:**
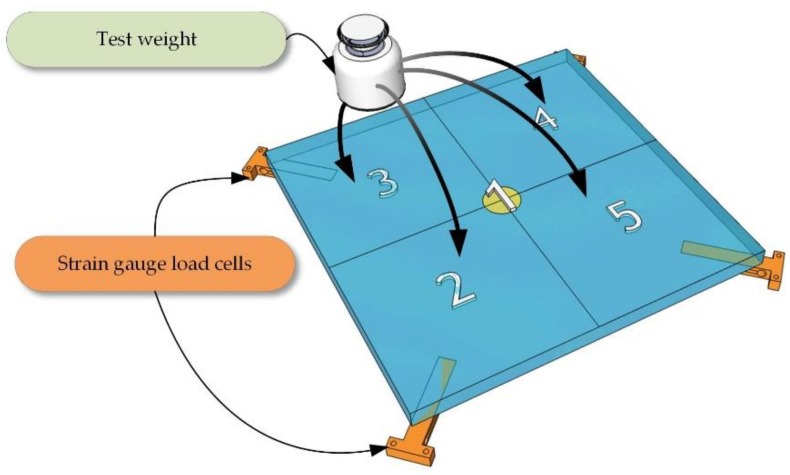
The procedure of test weight placement during the calibration process of weighing instruments.

**Figure 3 sensors-18-04200-f003:**
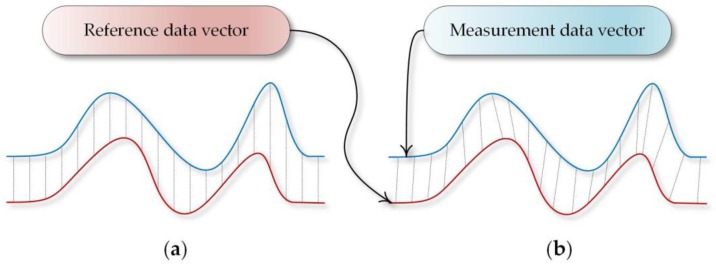
Visualization of shape matching with the use of element-wise Euclidean vector distance (**a**) and Dynamic Time Waring mapping (**b**).

**Figure 4 sensors-18-04200-f004:**
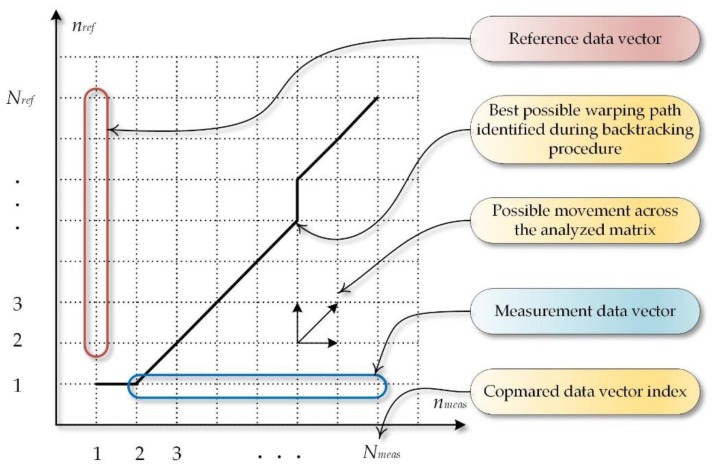
Visualization of the basic data structure and the operations exploited in the DTW (Dynamic Time Warping) algorithm.

**Figure 5 sensors-18-04200-f005:**
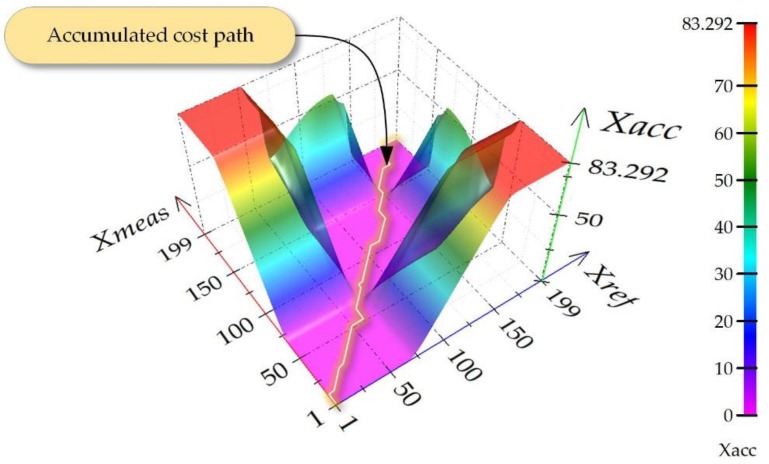
Visualization of a data matrix containing a so called cumulated cost and the least value path.

**Figure 6 sensors-18-04200-f006:**
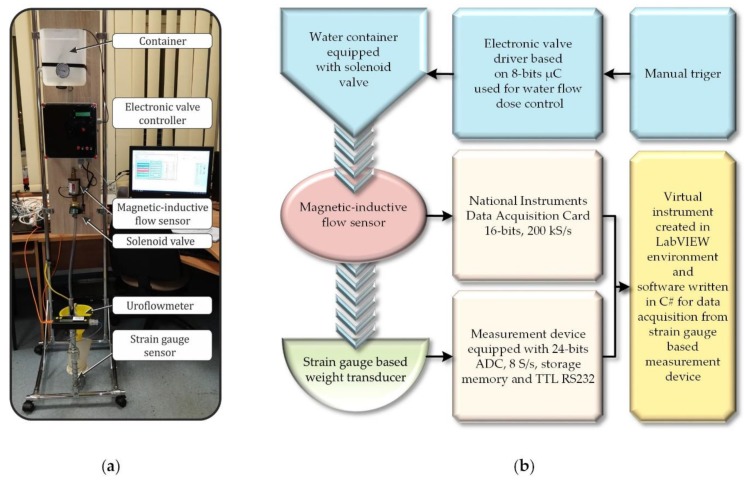
General view of the laboratory stand (**a**) and block diagram of the main elements used for the flow analysis of the tested uroflowmeter (**b**) [17].

**Figure 7 sensors-18-04200-f007:**
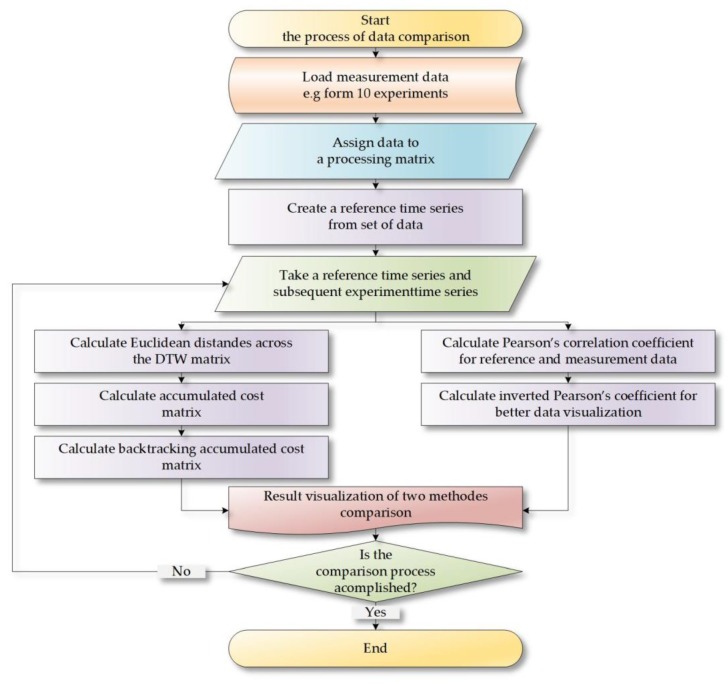
The algorithm of the procedure in which a magneto-inductive flowmeter and a uroflowmeter were compared.

**Figure 8 sensors-18-04200-f008:**
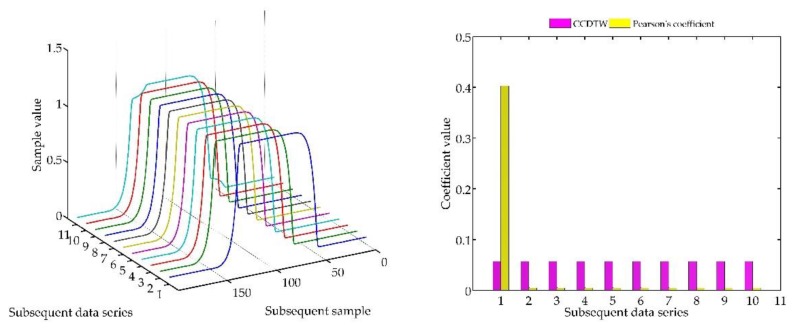
Simulation results achieved for data shifted by −15 samples for the first series (blue line) for *r_ampl_* = 2.

**Figure 9 sensors-18-04200-f009:**
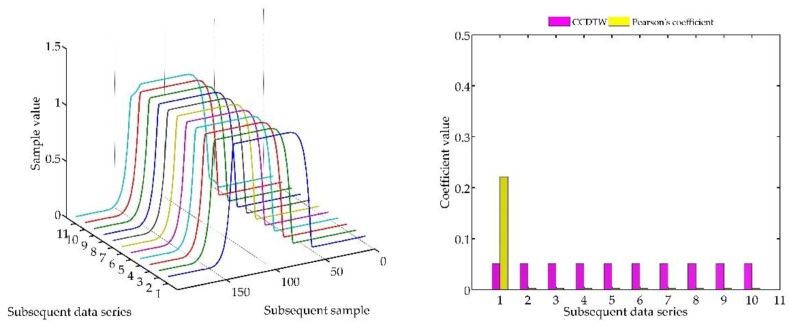
Simulation results achieved for data shifted by −10 samples for the first series (blue line) for *r_ampl_* = 2.

**Figure 10 sensors-18-04200-f010:**
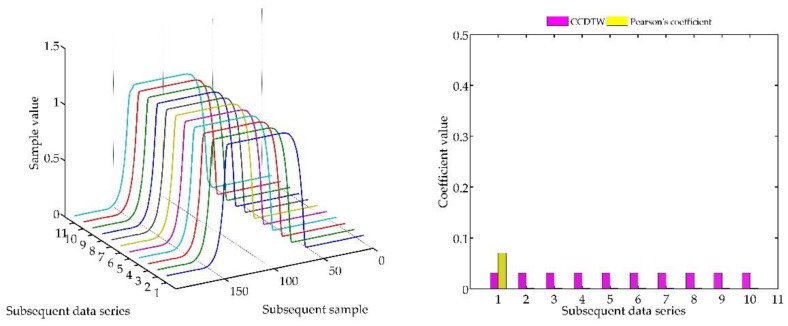
Simulation results achieved for data shifted by −5 samples for the first series (blue line) for *r_ampl_* = 2.

**Figure 11 sensors-18-04200-f011:**
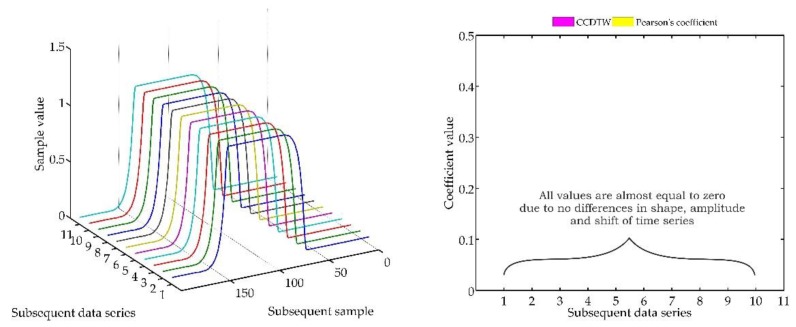
Simulation results achieved for non-shifted data for *r_ampl_* = 2.

**Figure 12 sensors-18-04200-f012:**
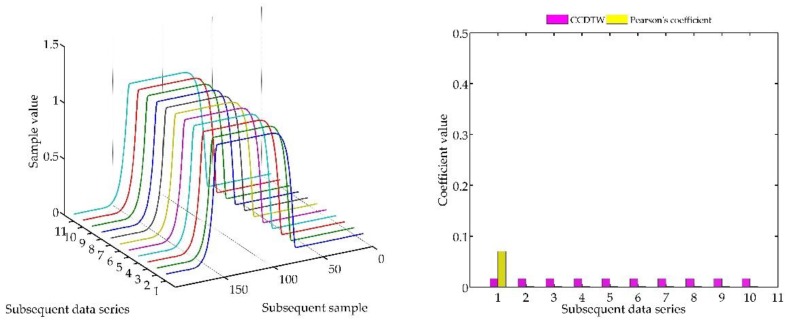
Simulation results achieved for data shifted by +5 samples for the first series (blue line) for *r_ampl_* = 2.

**Figure 13 sensors-18-04200-f013:**
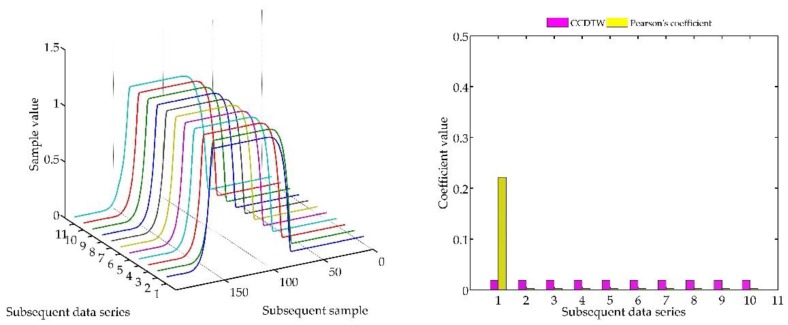
Simulation results achieved for data shifted by +10 samples for the first series (blue line) for *r_ampl_* = 2.

**Figure 14 sensors-18-04200-f014:**
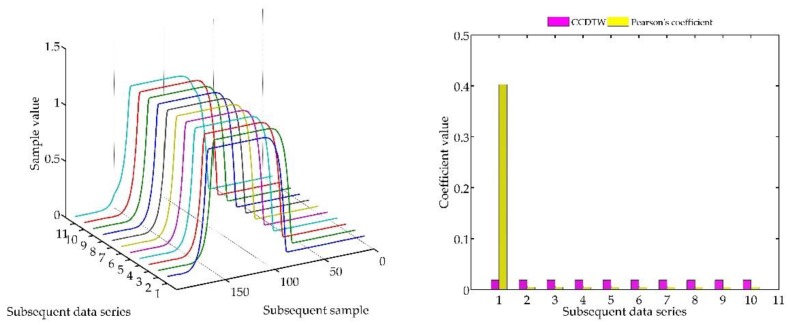
Simulation results achieved for data shifted by +15 samples for the first series (blue line) for *r_ampl_* = 2.

**Figure 15 sensors-18-04200-f015:**
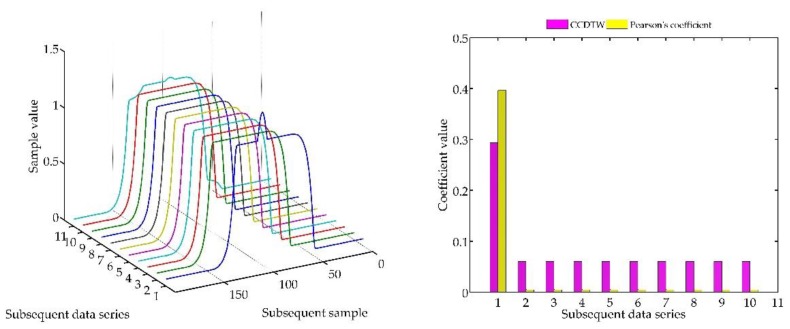
Simulation results achieved for data shifted by −15 samples for the first series (blue line) for *r_ampl_* = 2.

**Figure 16 sensors-18-04200-f016:**
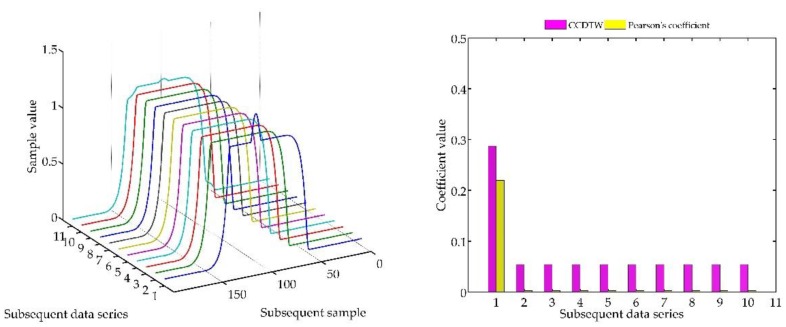
Simulation results achieved for data shifted by −10 samples for the first series (blue line) for *r_ampl_* = 2.

**Figure 17 sensors-18-04200-f017:**
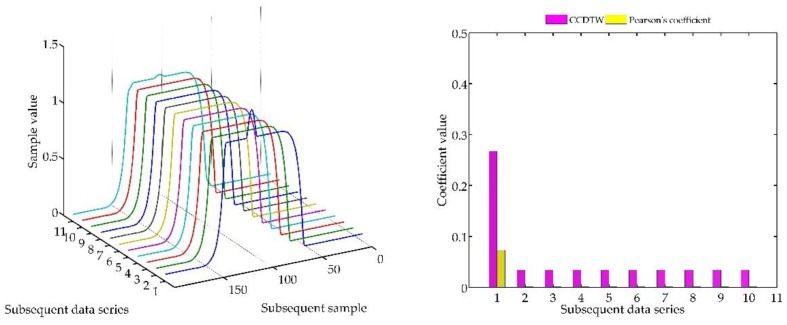
Simulation results achieved for data shifted by −5 samples for the first series (blue line) for *r_ampl_* = 2.

**Figure 18 sensors-18-04200-f018:**
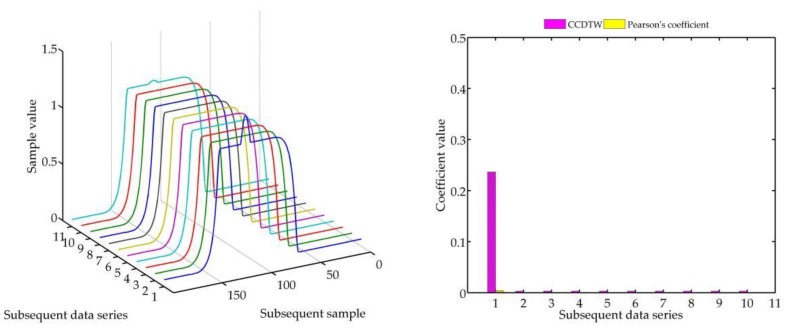
Simulation results achieved for non-shifted data for *r_ampl_* = 2.

**Figure 19 sensors-18-04200-f019:**
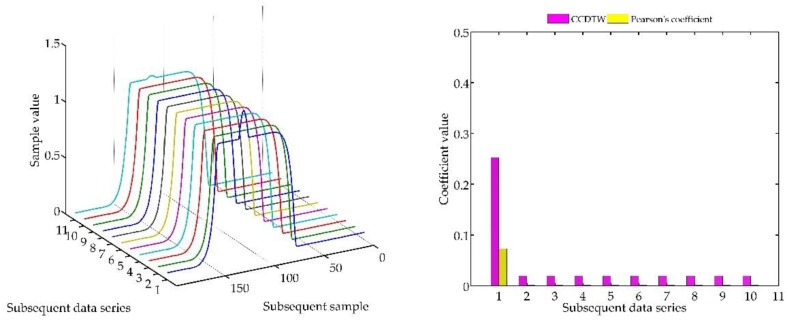
Simulation results achieved for data shifted by +5 samples for the first series (blue line) for *r_ampl_* = 2.

**Figure 20 sensors-18-04200-f020:**
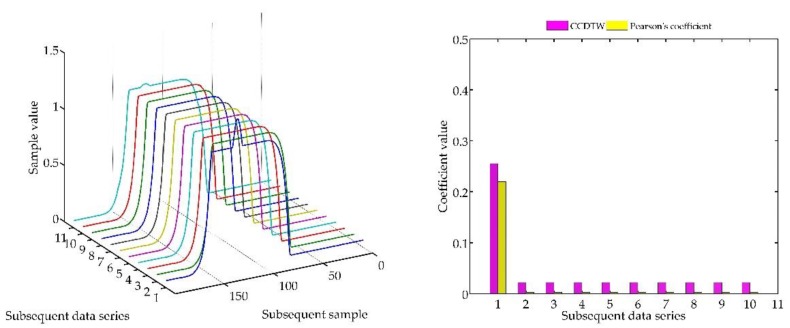
Simulation results achieved for data shifted by +10 samples for the first series (blue line) for *r_ampl_* = 2.

**Figure 21 sensors-18-04200-f021:**
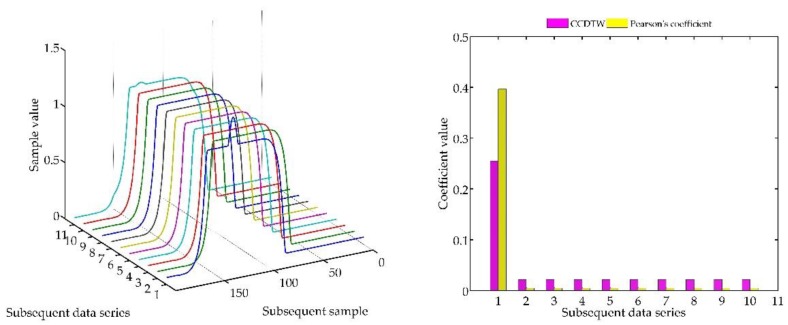
Simulation results achieved for data shifted by +15 samples for the first series (blue line) for *r_ampl_* = 2.

**Figure 22 sensors-18-04200-f022:**
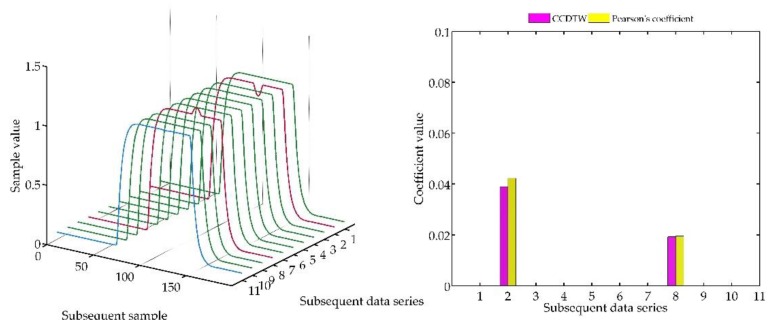
Simulation results obtained for disturbed charts № 2 and № 8 for *r_ampl_* = 100.

**Figure 23 sensors-18-04200-f023:**
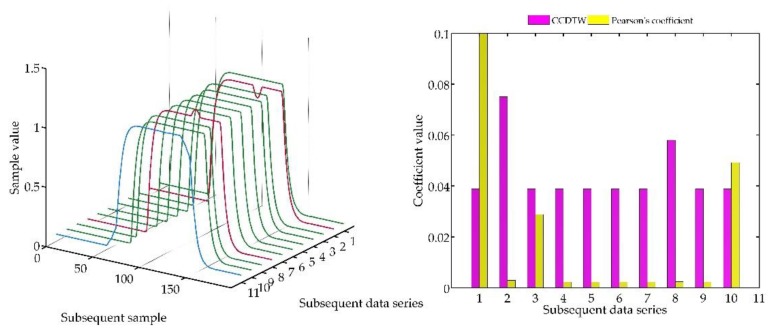
Simulation results obtained for disturbed non shifted charts № 2, № 8, and shifted non disturbed charts № 1, № 3, and № 10 for *r_ampl_* = 1.

**Figure 24 sensors-18-04200-f024:**
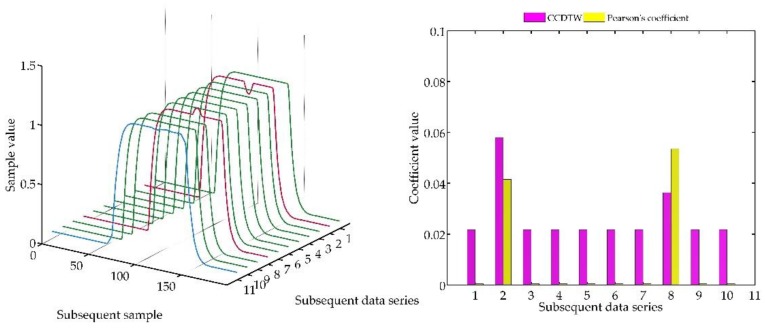
Simulation results obtained for disturbed and shifted charts № 2 and № 8 for *r_ampl_* = 1.

**Figure 25 sensors-18-04200-f025:**
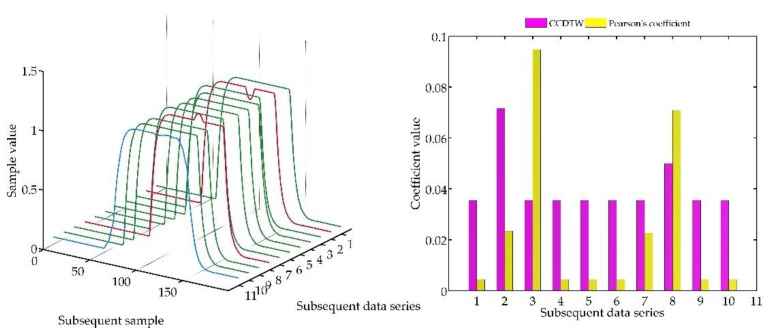
Simulation results obtained for shifted charts № 3 and № 7, and additionally disturbed № 2 and № 8 for *r_ampl_* = 1.

**Figure 26 sensors-18-04200-f026:**
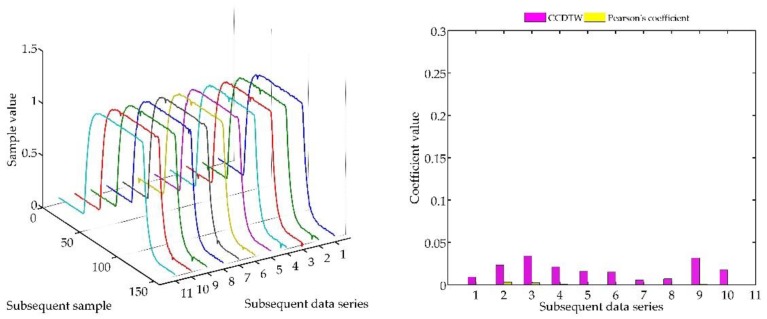
Data comparison results achieved in the measurement experiment for *r_ampl_* = 2.

**Figure 27 sensors-18-04200-f027:**
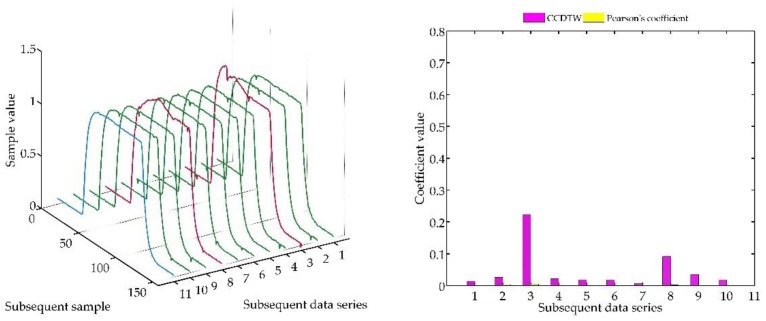
Simulation results based on measurement data for artificially disturbed charts № 3 and № 8 for *r_ampl_* = 2.

**Figure 28 sensors-18-04200-f028:**
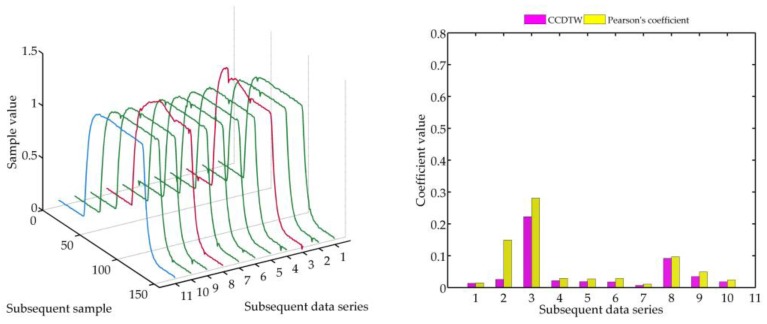
Simulation results based on measurement data for artificially disturbed charts № 3 and № 8 for *r_ampl_* = 100.

**Figure 29 sensors-18-04200-f029:**
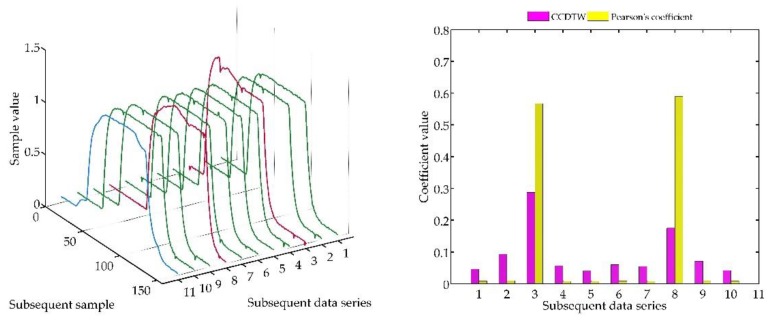
Simulation results based on measurement data for artificially disturbed and shifted charts № 3 and № 8 for *r_ampl_* = 2.

**Figure 30 sensors-18-04200-f030:**
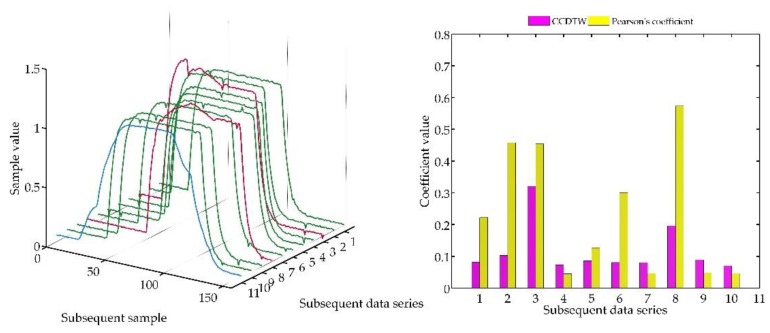
Simulation results based on measurement data for artificially disturbed, shifted charts № 3, № 8, and shifted non disturbed charts № 1, № 2, № 5, and № 6 for *r_ampl_* = 2.

**Table 1 sensors-18-04200-t001:** Magnetic-inductive flow meter parameters used in the comparison experiment [40].

Parameter	Value
Measuring range	0.10 ÷ 25.00 L/min
Accuracy	±(2% MW + 0.5% MEW)
Resolution	0.05 L/min
Repeatability	±0.2% MEW
Analog output	4 ÷ 20 mA, 0 ÷ 10 V

where: MW—measured value and MEW—final value of the measuring range.

**Table 2 sensors-18-04200-t002:** The uncertainty budget of the uroflowmeter used in the experiment [36].

Calculated Parameter	Value
Limiting error of mass measurement in ambient temperature	0.500 g
Standard uncertainty (type B)	0.290 g
Combined uncertainty of flow measurement	4.080 g/s
Coverage factor *kp*	2.327
Expanded uncertainty of flow measurement	9.500 g/s

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
