# Peer review of "Dynamic Outlier Detection in the Calibration by Comparison Method Applied to Strain Gauge Weight Sensors"

_sensors, 2018, doi:10.3390/s18124200_

Round 1

Reviewer 1 Report

Rows 81-84 - it not clear how the mass is placed on the device. Maybe a drawing will be more suggestive

Row 99 - “results of comparison of” - one “of” too much

Fig. 4 - Yacc instead of Xacc?

Eq7 - Who is Y?

Eq8 - Xcum - the index "cum", may be it can be renamed.

Fig 10 - the graph on the right is empty? May be the values 0 can be shown somehow.

Author Response

Dear Reviewer,

Thank you for your valuable comments. The article was changed according to all reviewer’s recommendation. Three additional pages were added. Moreover, an external native speaker expert suggested some additional changes and description enrichment.

The main changes which were made can be listed as follows:

 the title of the article is changed from:

Dynamic Outlier Detection in the Calibration by Comparison Method Applied to the Strain Gauge Weight Sensors, to

Dynamic Outlier Detection in the Calibration by Comparison Method Applied to Strain Gauge Weight Sensors – the article “the” was removed;

 minor changes in the abstract, the main text and discussion sections;

a new Figure 2. was attached in document;

 the error (missed last point Nref, Nmeas) in Figure 4 was corrected;

 the error in equation 7 was corrected;

 Figure 11 were modified by adding some information;

 the quality of all images was improved;

the new subsection 5.3. entitled Simulation results of comparing the Pearson's correlation coefficient method and the Calibration-by-Comparison Dynamic Time Warping for an ideal time series with two disturbed series,  was included;

 additional description of the experiments results were added;

 the article language correction was made.

The full review response you can find in the attached file.

Reviewer 2 Report

 The formula “k ≥ 5“ in line 77 should be rendered in italics.

I recommend that the author significantly improve the resolution of the images (in Fig. 4, for example), using lossless compression (the current version of the manuscript contains compression artifacts around the axes of the graphs and their decriptions). Graphs 7 to 20 generally exhibit inferior quality, with the grids either lacking or blurred due to low resolution or inadequate compression. 

As regards the formal aspects, the manuscript is written clearly, in relatively good English; however, I would still recommend some revision of the style (expressions) and the grammar (for instance, lines 33 -  approved for the use, e.g. for everyday life…; 49 - comparative tests of time characteristics of…; 119 – in majority of…).

Author Response

Dear Reviewer,

Thank you for your valuable comments. The article was changed according to all reviewer’s recommendation. Three additional pages were added. Moreover, an external native speaker expert suggested some additional changes and description enrichment.

The main changes which were made can be listed as follows:

 the title of the article is changed from:

Dynamic Outlier Detection in the Calibration by Comparison Method Applied to the Strain Gauge Weight Sensors, to

Dynamic Outlier Detection in the Calibration by Comparison Method Applied to Strain Gauge Weight Sensors – the article “the” was removed;

 minor changes in the abstract, the main text and discussion sections;

a new Figure 2. was attached in document;

 the error (missed last point NrefNmeas) in Figure 4 was corrected;

 the error in equation 7 was corrected;

 Figure 11 were modified by adding some information;

 the quality of all images was improved;

the new subsection 5.3. entitled Simulation results of comparing the Pearson's correlation coefficient method and the Calibration-by-Comparison Dynamic Time Warping for an ideal time series with two disturbed series,  was included;

 additional description of the experiments results were added;

 the article language correction was made.

The full review response you can find in the attached file.

Reviewer 3 Report

The paper proposes the use of CCDTW (Comparison/Calibration Dynamic Time Warping) for sensor calibration. CCDTW uses the Dynamic Time Warping algorithm (DTW) for comparing a set of series of experiments with their reference (which is, in this case, the average time series). Series which are considerably different from the reference may be considered outliers, and removed from reference set. Furthermore, CCDTW is less sensitive to shifts in sample position. The approach is interesting, and according to the authors, CCDTW has not been used in this context. However, I’d like a more thoroughly empirical validation, with more than one disturbed or shifted series in the artificial data sets. Furthermore, the experimental analysis could be improved, with a better description of the figures 7 to 24. I did not understand how r_ampl was determined in the experiments. Last but not least, although the author proposes as future work the extension to other types of the sensor, I think that the if it is possible to include another sensor in the present analysis will better show the suitability of the method.

Author Response

(The authors gave the same response as above.)

Round 2

Reviewer 3 Report

Authors have addressed may main concerns